# Effect of Trilobatin from *Lithocarpus polystachyus* Rehd on Gut Microbiota of Obese Rats Induced by a High-Fat Diet

**DOI:** 10.3390/nu13030891

**Published:** 2021-03-10

**Authors:** Hailiang Shen, Linhua Huang, Huating Dou, Yali Yang, Houjiu Wu

**Affiliations:** 1Citrus Research Institute, Southwest University, Chongqing 400000, China; shl2021@email.swu.edu.cn (H.S.); huanglh@cric.cn (L.H.); douhuating@cric.cn (H.D.); 2Citrus Research Institute, Chinese Academy of Agricultural Science, Chongqing 400000, China; 3Department of Food Engineering and Nutritional Science, Shaanxi Normal University, Xi’an 710000, China; yangyali@snnu.edu.cn; 4National Research and Development Center of Apple Processing Technology, Xi’an 710000, China

**Keywords:** antiobesity, trilobatin, *Lithocarpus polystachyus* Rehd, gut microbiota, KEGG pathway

## Abstract

Trilobatin was identified as the primary bioactive component in the *Lithocarpus polystachyus* Rehd (LPR) leaves. This study explored the antiobesity effect of trilobatin from LPR leaves and its influence on gut microbiota in obese rats. Results showed that trilobatin could significantly reduce body and liver weight gain induced by a high-fat diet, and the accumulation of perirenal fat, epididymal fat, and brown fat of SD (Male Sprague–Dawley) obese rats in a dose-independent manner. Short-chain fatty acids (SCFAs) concentrations increased, especially the concentration of butyrate. Trilobatin supplementation could significantly increase the relative abundance of *Lactobacillus*, *Prevotella*, *CF231*, *Bacteroides*, and *Oscillospira*, and decrease greatly the abundance of *Blautia*, *Allobaculum, Phascolarctobacterium*, and *Coprococcus*, resulting in an increase of the ratio of *Bacteroidetes* to *Firmicutes* (except the genera of *Lactobacillus* and *Oscillospira*). The Kyoto Encyclopedia of Genes and Genomes (KEGG) pathway predicted by the Phylogenetic Investigation of Communities by Reconstruction of Unobserved States (PICRUSt) indicated the different relative metabolic pathways after trilobatin supplementation. This study may reveal the contribution of gut microbiota to the antiobesity effect of trilobatin from LPR leaves and predict the potential regulatory mechanism for obesity induced by a high-fat diet.

## 1. Introduction

Obesity has become a global epidemic and about 25 million people die every year because of overweight and obesity [1,2]. The World Health Organization regards obesity as abnormal and excessive fat accumulation, which could impair health. Studies indicated that obesity is a fat metabolism disorder, which not only affects body image, but also is associated with many diseases including type II diabetes, mellitus, coronary heart disease, fatty liver, hypertension, and so forth. [3,4]. The prevention and treatment of obesity is critical to reducing the incidence of morbidity and mortality. Several mechanisms are proposed to prevent obesity, which include the decrease of energy and food intake and lipogenesis, and the increase of energy expenditure, lipolysis, and fat oxidation [5]. Many traditional antiobesity strategies have been conducted, such as dieting, strenuous exercise, drug therapy, and so on. However, dieting and strenuous exercise usually require long-term persistence, while drug therapy for antiobesity consists of appetite suppressants which regulate neurotransmitters in the hypothalamus, but it has many limitations including the side effects, high recurrence rate, and so forth [6]. Thus, finding natural plant extracts and compounds to cure obesity appears a more relevant strategy with less adverse effects.

*Lithocarpus polystachyus* Rehd (LPR), belonging to the genus Fagaceae, is an evergreen tree [7]. The leaves of LPR are known to taste sweet and are usually used as a kind of sweet tea in the regions of the Yangtze River of China, as well as in Southeast Asia, such as India, Thailand, Laos, and so forth. [8,9]. The leaves of LPR have been allowed as new food materials in China since 2017 because of their medicinal and edible functions, as they contain nonclassical flavonoids, dihydrochalcones, such as phlorizin/phloretin, and trilobatin [10,11,12]. Trilobatin and phlorizin were identified as the primary bioactive components in the leaves of LPR. It is well known that phlorizin is abundant in apple peels and its content in LPR leaves is more than 100 times that in apples. Shin et al. [13] found that phlorizin could decrease body weight in obese animal models and reduce insulin resistance. However, the content of phlorizin is lower than that of trilobatin in LPR leaves. The main health benefits of LPR leaves are antioxidant, anti-inflammation, antimicrobial, antidiabetic, anticancer, hepatoprotection, cardioprotection, antiobesity, neuroprotection, and antiallergic activities [10]. Although some studies showed that the leaves of LPR could decrease body weight gain of obese rats caused by a high-fat diet [14], few studies have reported that trilobatin, the main flavonoid compound in LPR leaves, exerts an antiobesity effect in animal models.

The flavonoid compounds from plant extracts have shown many beneficial effects in humans and animals. The beneficial effects in vivo may be mediated through interactions with the gastrointestinal tract including inhibiting enzymes and modulating gut barrier properties and enteroendocrine secretion [15]. The gut microbiota participates in the regulation of lipid metabolism. Several reports found that obesity is related to gut microbiota dysbiosis [16]. The eight-week administration of cranberry extract, rich in flavonoids, exerted antiobesity effects by modulating the gut microbiota including the increase of the proportion of the mucin-degrading bacterium *Akkermansia* [17], while antiobesity effects of Fubrick teas on the gut microbiota were through increasing the relative abundance of *Bacteroides*, *Adlercreutzia*, *Alistipes*, and *Parabacteroides* and decreasing that of *Staphylococcus* [18]. However, the effect of trilobatin from LPR leaves on the gut microbiota and the regulatory function of gut microbiota in lipid metabolism remain insufficiently revealed.

Therefore, we studied the antiobesity mechanism of trilobatin from LPR leaves at the gastrointestinal tract level. We hypothesized that the modulation of gut microbiota could exert antiobesity effects of trilobatin from LPR leaves on metabolic health. 

## 2. Materials and Methods

### 2.1. Materials

Trilobatin from LPR leaves was obtained from Shanghai Guchen Biotechnology Co., Ltd. (Shanghai, China). Orlistat was purchased from Zhongshan Wanhan Pharmaceutical Co. Ltd. (Guangdong, China). QIAamp DNA Stool Mini Kit was purchased from QIAGEN (Frankfurt, Germany).

### 2.2. Animals and Diet

SD rats of 200 ± 10 g and 4–6 weeks old were purchased from Chengdu Dossy Experimental Animals Co., Ltd. (Chengdu, China). The experimental protocol was approved by the Experimental Animal Ethics Committee of West China Hospital of Sichuan University (no: 2020328A). SD rats were housed under standard environment conditions with free access to food and distilled water. Both chow diet and high-fat diet were obtained from Chengdu Dossy Biotechnology Co., Ltd. (Chengdu, China). Appendix A shows the composition of the chow diet, and the high-fat diet contained 70% chow diet, 15% lard, 10% egg yolk, and 5% cholesterol. 

After SD rats were maintained under standard conditions for five days, 135 rats were randomly divided into two groups which were the normal control group (*n* = 15) and the high-fat diet group (*n* = 120). The former group was fed a low-fat diet, and the latter group a high-fat diet. After 30 days, the rats were weighed. When the weight of the high-fat diet group was significantly higher than the average weight of the normal control group (*p* < 0.05), the obesity model was obtained. Then, the SD rats were divided into six groups (*n* = 6 for each group): a low-fat diet group as normal control (NC); a high-fat diet group as blank control (BC); a high-fat diet with orlistat group as positive control (ORL); a high-fat diet with low dose of trilobatin group (TRL); a high-fat diet with middle dose of trilobatin group (TRM); a high-fat diet with high dose of trilobatin group (TRH). SD rats in the TRL, TRM, and TRH groups received 30 mg/kg, 60 mg/kg, and 120 mg/kg of body weight through intragastric gavage, respectively, while SD rats in the ORL group were treated with orlistat of 3.75 mg/kg of body weight (BW), and all groups were treated with the same volume of saline for four weeks. During the experiments, the food intake and body weight were recorded every three days. Two days before the end of treatment, the feces of the SD rats in all tested groups were taken by aseptic centrifuge tubes and stored at −80 °C. After four weeks of treatment, the SD rats were fasted overnight for 12 h, and then sacrificed. The adipose tissues, livers, and cecal contents were removed and weighed, and then stored at −80 °C for further analysis. 

### 2.3. Short-Chain Fatty Acid Analysis

According to the SCFAs analysis reported by Wu et al. [19], to extract SCFAs, 0.1 g of cecal contents were dissolved with pure water and centrifuged at 4500× *g* for 10 min. Next, 2.0 mL of supernatant, 0.2 mL of 50% sulfuric acid solution, and 2 mL of anhydrous ether were mixed and placed for 30 min at 4 °C. Then, the supernatant was filtered through 0.45 μm filter membrane and used for analysis by a gas chromatograph (GC). 

SCFAs in the cecal contents were determined by a gas chromatograph (Agilent 7890B, California, CA, USA) equipped with flame ionization detector and a gas chromatography column (HP-FFAP, 60–240 °C, 30 m × 320 μm, China). Gas chromatography conditions: The initial oven temperature was 80 °C for 5 min, 5 °C/min to 250 °C for 5 min. Nitrogen was the carrier gas at 12.3 mL/min and a split ratio of 1:10. The temperature of injection and flame ionization detector were 240 °C and 300 °C, respectively. The injection volume was 1 μL.

### 2.4. Gut Microbiota Analysis

The total DNA was isolated from the frozen feces with an QIAamp DNA stool mini kit (Qiagen, Frankfurt, Germany) according to the kit instructions. The quality of total DNA was accessed by 1.2% agarose gel electrophoresis, and was cut off the target strip. The purified products were recovered by the gel recovery kit (Qiagen, Frankfurt, Germany). The sequencing analysis of purified PCR products was performed by the Chengdu Dossy Experimental Animals Co., Ltd. (Chengdu, China) using illumina platform. 

The alpha diversity was analyzed by the Chao index, Simpson index, Shannon diversity index, and Goods coverage. Beta diversity was evaluated using principal coordinate analysis (PCoA) at the OUT level. The abundance of gut microbiota at the genus level was identified by principal component analysis (PCA), linear discriminant analysis (LDA), effect size (LEfSe), and orthogonal partial least squares discrimination analysis (OPLS-DA).

### 2.5. Predictive Metagenome Analysis

The PICRUSt was used to predict the metagenome, based on the 16S amplicon data [20]. The predicted metagenomes were annotated functionally by KEGG pathways, and the functional prediction was exported as KEGG orthology (KO) levels. The present data included the level 1 and 2 function.

### 2.6. Statistical Analysis

Data were expressed as means ± standard deviations. The results were analyzed by SPSS software (version 20, New York, NY, USA). The Kolmogorov–Smirnov test was used to check the normal distribution of the data. Significance differences between the tested groups were conducted by Dunnett post hoc test after a significant one-way ANOVA. Statistical significance was regarded at *P* < 0.05.

## 3. Results

### 3.1. Food Intake, Morphometric Parameters, and SCFAs Are Greatly Modulated by Trilobatin from LPR Leaves

Table 1 shows the changes in food intake, morphometric parameters including body weight, liver weight, perirenal adipose tissue (PAT), epididymal adipose tissue (EAT), brown adipose tissue (BAT), and total adipose tissue (sum AT), and SCFAs of different experimental groups. In comparison with the NC group, the BC group showed significant increases in the body weight gain, liver weight, PAT, EAT, BAT, and sum AT weight of rats (*p* < 0.05). When the obese rats were treated with different dosages (30 mg/kg, 60 mg/kg, and 120 mg/kg BW) of trilobatin from LPR leaves and orlistat, the data of all the above were less than those of the high-fat diet group. These results exhibited that the trilobatin from LPR leaves in the experimental concentrations, especially 60 mg/kg and 120 mg/kg of trilobatin, greatly decreased the body weight and liver weight gain, and the content of perirenal fat, epididymal fat, and brown fat of obese rats (*p* < 0.05), and the protective effects were dose-independent. The food intake of SD rats was recorded during the experimental period and no significant differences were between each group (*p > 0.05*).

SCFAs (acetic, propionic, butyric, isobutyric, valeric, and isovaleric acids) in the cecal content of SD rats were measured using gas chromatography. Compared with the NC group, the content of acetic acid and butyric acid significantly decreased (*p* < 0.05) in the BC group. However, compared with the BC group, the level of propionic acid significantly increased in the middle dose of trilobatin group (*p* < 0.05), while the level of butyric acid greatly increased in the high dose of trilobatin group (*p* < 0.05).

### 3.2. Effects of Trilobatin on the Diversity and Structure of Gut Microbiota

A total of 945,104 raw reads were collected from 36 samples among six groups (*n* = 6) with an average of 157,517 reads for each group. The calculated microbial community alpha diversity indexes (Shannon, Chao1, Simpson, observed OTUs, and Goods coverage) are shown in Figure 1. The coverage indexes of all tested groups were more than 98% with no significant difference. Compared to both NC and BC groups, the Chao 1 and Shannon index and observed OTUs of the ORL, TRL, TRM, and TRH groups were lower (*p* < 0.05). However, there were no significant differences between the NC and BC groups (*p* > 0.05). The results indicated that the high-fat diet did not affect the alpha diversity of the microbial community of SD rats, whereas the trilobatin and orlistat treatment significantly reduced the diversity (*p* < 0.05). 

The composition of gut microbiota in the orlistat and trilobatin administration groups revealed by PCoA was dramatically distinct from the NC (*p* < 0.01) and BC (*p* < 0.05) groups using both weighted and unweighted UniFrac distance (Figure 2A,B). A total of 34 bacteria phyla were identified in all tested samples and *Firmicutes* and *Bacteroidetes* were the two most dominant phyla, accounting for 92.76%, followed by *Spirochaetes* (3.05%) and *Proteobacteria* (2.05%) (Figure 3A). Compared to the BC group, the trilobatin administration greatly reduced the relative abundance of *Firmicutes* (*p* < 0.05) and increased that of Bacteroidetes (*p* < 0.05), which revealed that the trilobatin could dramatically inhibit the increase of the ratio of *Firmicutes* to *Bacteroidetes* (*F/B*) induced by the high-fat diet. Figure 3B shows the relative abundances of the top 20 abundant genera. The most dominant genera in all tested groups were *Lactobacillus, Blautia*, and *Prevotella*. Compared with the NC group, the relative abundance of *Lactobacillus, Prevotella, CF231, Bacteroides*, *and Oscillospira* decreased remarkably (*p* < 0.01), whereas that of *Blautia, Allobaculum, Phascolarctobacterium, Coprococcus*, *and Sutterella* increased significantly (*p* < 0.01) in the BC group (Figure 3C–M). Additionally, compared with the BC group, the trilobatin supplementation dramatically increased the relative abundance of *Lactobacillus, Prevotella, CF231, Bacteroides*, *and Oscillospira* (*p* < 0.01), but decreased that of *Blautia, Allobaculum, Phascolarctobacterium*, *and Coprococcus* (*p* < 0.01). However, the effect of trilobatin on the relative abundance of abundant genera of obese rats was dose-independent, especially the *Sutterella*. The relative abundance of *Sutterella* in the TRL and TRM group dramatically decreased the high-fat-diet-induced increase (*p* < 0.01), while the high-dose trilobatin treatment caused an increase of the relative abundance of *Sutterella* (*p* < 0.01).

The abundant bacterial genera (top 20 genera) were selected to obtain a heat map revealing an intuitionistic relative abundance and the difference of abundance (Figure 4A). All the top 20 abundant genera, except *Sutterella,* belong to *Bacteroidetes* and *Firmicutes*. There were more than 86.23% reads of uncultured bacteria of all samples at the species level and the species analysis was not further conducted. PCA and OPLS-DA analysis showed the relationship among the relative abundance of gut microbiota in the tested groups (Figure 4B,C). The NC group plotted was far away from the BC group, while the ORL, TRL, TRM, and TRH groups plotted were between the NC and BC groups. These results indicated that trilobatin administration could change the gut microbiota communities caused by a high-fat diet.

### 3.3. Functional Prediction Analysis

The higher relative abundance of KEGG pathways predicted by PICRUSt in all groups were carbohydrate metabolism, amino acid metabolism, cofactors and vitamins metabolism, metabolism of terpenoids and polyketides, replication and repair, other amino acids metabolism, lipid and energy metabolisms (Figure 5A). To further predict the metabolic pathways, we have done the second level of KEGG pathway analysis predicted by PICRUSt among all tested groups. Thirteen obviously different KEGG metabolic pathways in the NC and BC groups, three different metabolic pathways in the BC and TRL groups, four different metabolic pathways in the BC and TRM groups, and five different metabolic pathways in the BC and TRH groups were imported into KEGG analysis (Appendix A). Based on VIP > 1 and *p* value < 0.05, nine enriched and relative metabolic pathways predicted by PICRUSt (steroid biosynthesis, steroid hormone biosynthesis, geraniol degradation, D-arginine and D-ornithine metabolism, linoleic acid metabolism, retinol metabolism, carotenoid biosynthesis, flavonoid biosynthesis, and tropane, piperidine, and pyridine alkaloid biosynthesis) were selected to evaluate the correlations between serum metabolites and gut microbiota. As shown in Figure 5B, *Collinsella* was negatively correlated with steroid hormone biosynthesis (*p* < 0.05, *r* = −0.82). *Roseburia* was positively correlated with geraniol degradation (*p* < 0.001, *r* = 0.99). D-arginine and D-ornithine metabolism were positively associated with *Blautia* (*p* < 0.01, *r* = 0.94) and *Phascolarctobacterium* (*p* < 0.01, *r* = 0.93), but negatively correlated with *Oscillospira* (*p* < 0.05, *r* = −0.83), *Bacteroides* (*p* < 0.05, *r* = −0.89), and *YRC22* (*p* < 0.05, *r* = −0.87). There was negative and significant correlation between *Allobaculum* and linoleic acid metabolism (*p* < 0.05, *r* = −0.83). Retinol metabolism was negatively correlated with *Faecalibacterium* (*p* < 0.05, *r* = −0.89), but positively correlated with *CF231* (*p < 0.01, r = 0.94*) and *Bacteroides* (*p* < 0.01, *r* = 0.93). *Bacteroides* was negatively correlated with carotenoid biosynthesis (*p* < 0.05, *r* = −0.83), while *Roseburia* was negatively and significantly correlated with flavonoid biosynthesis (*p* < 0.01, *r* = −0.92).

## 4. Discussion

Obesity is an important topic for health [2]. A high-fat diet in the long term may promote energy intake and result in the increasing of visceral fat deposition, considered to be a main factor of obesity [21]. LPR is widely in the mountainous region of southern China and its leaves have been regarded as a traditional Chinese sweet tea for many years [22]. The leaves of LPR could be used for the treatment of disorders such as diabetes, hypertension, epilepsy, and antiobesity, as well as improving leptin resistance in diet-induced obese rats [14,23,24]. Three dihydrochalcones, including trilobatin, phloridzin, and phloretin, were regarded as the major flavonoid compounds in the leaves of LPR. Several reports revealed the flavonoids of LPR leaves, especially phlorizin and phloretin, have anti-inflammatory, antioxidant, antidiabetic, and antiobesity functions [25]. However, no research stated the trilobatin isolated from LPR leaves was the primary contributor to the antiobesity effect. In the present study, we hypothesized that the trilobatin from LPR leaves has a significant antiobesity effect on SD obese rats. In our study, we evaluated the antiobesity influence of trilobatin on SD obese rats and investigated the underlying mechanism for this effect. 

The four-week administration of trilobatin isolated from LPR leaves in the low-, middle-, and high-concentration groups could significantly reduce the body and liver weight gain of SD obese rats (*p* < 0.05), confirming that the trilobatin of LPR leaves had functions of preventing obesity, which was consistent with previous research for the extracts of LPR leaves [14,23]. Compared to the BC group, the food intake during the four-week trilobatin treatment was not significantly different (*p* > 0.05), which indicated the trilobatin-induced weight loss of obese rats may be due to the abnormal absorption of foods or the decrease of body energy storage in vivo [14,26].

The undigested carbohydrates were rapidly fermented to SCFAs in the gut microbiota environment. The amounts and profiles of SCFAs could be affected by the dietary properties [27]. Acetate was an important substrate in the synthesis of liver cholesterol and fatty acid, and it increases colonic blood flow, oxygen uptake, and ileal motility [28]. Propionate is a precursor in the liver during the synthesis of lipid and protein [29], while butyrate is an energy source for colonocytes [28]. Previous studies showed that trilobatin from LPR leaves could reverse the decrease of SCFA concentrations caused by a high-fat diet, especially the amount of butyrate treated with a high dose of trilobatin. The high-fat diet increased pH value in the cecal contents, whereas addition of trilobatin decreased the cecal pH to normal levels, which was consistent with other research for antiobesity effects [30]. 

The results of gut microbiota analysis suggested that lower alpha diversity indexes, especially Chao and Shannon indexes, were generated in the middle- and high-dose trilobatin groups than in the other groups, which may be due to the dominant bacterial communities restraining other populations [31]. The taxonomic analysis of microbiota communities indicated that *Firmicutes* and *Bacteroidetes* were the most dominant phyla in all groups, and this analysis agreed with previous research [32,33]. Our results further indicated that trilobatin treatment could change the gut microbiota profile and decrease *Firmicutes* and increase *Bacteroidetes* significantly compared to that in the BC group, consistent with the general view of the flavonoids effect [34]. Some studies stated that the ratio of *Firmicutes/Bacteroidetes* is a microbiome marker of obesity and type 2 diabetes in human and animal research [34], but a growing number of studies did not agree with that [32,35]. Thus, we should pay more attention to lower classification of gut microbiota to explain the mechanism of trilobatin supplementation effects.

The abundance of *Lactobacillus*, *Blautia*, and *Prevotella* was higher than that of other genera in all groups. *Prevotella*, *CF231* and *Bacteroides* belong to Bacteroidetes phylum, while *Blautia*, *Allobaculum*, *Phascolarctobacterium*, and *Coprococcus* are within the *Firmicutes* phylum. Compared to the genus of the high-fat diet group, trilobatin supplementation significantly increased the relative abundance of *Lactobacillus*, *Prevotella*, *CF231*, *Bacteroides*, and *Oscillospira* (*p* < 0.05), and greatly decreased the abundance of *Blautia*, *Allobaculum*, *Phascolarctobacterium*, and *Coprococcus* (*p* < 0.05), which was consistent with an increase of the ratio of *Bacteroidetes* to *Firmicutes* (except the genera of *Lactobacillus* and *Oscillospira*). *Lactobacillus* and *Oscillospira* belong to the *Firmicutes* phylum. It has been reported that the supplementation of fatty acids could enrich *Lactobacillus*, which could metabolize the fatty acids and alleviate inflammation to prevent liver disease [36]. Chen et al. [37] found a significant positive correlation between *Lactobacillus* and serum triglyceride. Interestingly, our results showed that trilobatin administration dramatically elevated the relative abundance of *Lactobacillus*, which may be associated with the blood lipid metabolism in the high-fat-diet rats. *Oscillospira* was positively associated with leanness and health [38]. The increase of relative abundance of *Oscillospira* after trilobatin treatment demonstrated that trilobatin from LPR leaves had significant antiobesity effect and no intestinal environment imbalance. *Oscillospira* is a large genus with high species diversity, which will require a finer-grained understanding of these species’ genetic potential and interactions with their host in further research. The above results confirmed that trilobatin from LPR leaves could regulate the structure and diversity of gut microbiota of SD obese rats.

Except for gut microbiota, we further investigate the effects of trilobatin treatment on the KEGG pathways predicted by PICRUSt. It was found that significantly different KEGG pathways predicted by PICRUSt between trilobatin administration (TRL, TRM, and TRH) and BC groups were mainly involved in lipid metabolism (steroid biosynthesis, steroid hormone biosynthesis), amino acid metabolism (D-arginine and D-ornithine metabolism), other secondary metabolites (flavonoid biosynthesis), terpenoid and polyketide (carotenoid biosynthesis), and cofactor and vitamin (retinol metabolism). Trilobatin from LPR leaves could be used as steroid-genesis modulators against 3β-hydroxysteroid dehydrogenase (HSD), 17β-HSD, and aromatase of the steroid-genesis pathway [39,40]. The steroid biosynthesis pathway has the ability to improve the squalene level, which could improve the oxidative stress and block lipid peroxidation [41,42]. Vitamins and amino acids are important for the fat synthesis from carbohydrate and protein [43], but the flavonoid biosynthesis and carotenoid biosynthesis pathways seem to have a role against obesity [44].

LPR is a traditional Chinese herb and its leaves have been taken as sweet tea for several hundred years without adverse effects or toxicity [45]. Trilobatin, a main flavonoid compound of LPR leaves, could significantly reduce body and liver weight gain and improve the gut microbiota communities of obese rats, which indicated trilobatin had potential functions for curing obesity induced by a high-fat diet, but it may have some limitations, including poor intestinal absorption and low bioavailability. Although there were few reports about trilobatin from LPR leaves for antiobesity, much research showed the antiobesity effects of phlorizin, whose structure is similar to trilobatin [25]. The present research suggested that the supplementation of trilobatin from LPR is a functional food for preventing obesity. In the future, investigations including biological activities of metabolites and toxicodynamics and its metabolites in food applications of trilobatin need to be studied.

## 5. Conclusions

In summary, trilobatin supplementation has a beneficial effect on high-fat-diet rats, including the body and liver weight gain. The gut microbiota analysis showed the middle and high dose of trilobatin reduced the alpha diversity indexes, especially the Chao and Shannon indexes. Compared to the genus of the high-fat diet group, trilobatin supplementation could significantly increase the relative abundance of *Lactobacillus, Prevotella, CF231, Bacteroides*, and *Oscillospira* (*p* < 0.05), and decrease greatly the abundance of *Blautia*, *Allobaculum, Phascolarctobacterium*, and *Coprococcus* (*p* < 0.05), resulting in an increase of the ratio of *Bacteroidetes* to *Firmicutes* (except the genera of *Lactobacillus* and *Oscillospira*). The KEGG analysis using PICRUSt predicted the different relative metabolic pathways after trilobatin supplementation. This study may provide a theoretical basis for the further study of the antiobesity effect of trilobatin from LPR leaves and predict the potential regulatory mechanism for obesity induced by a high-fat diet. 

## Figures and Tables

**Figure 1 nutrients-13-00891-f001:**
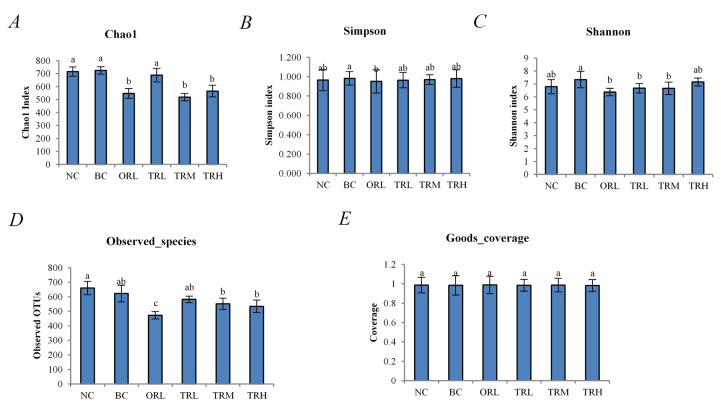
Alpha diversity of gut microbiota from different tested groups. (**A**) Chao1 index; (**B**) Simpson index; (**C**) Shannon index; (**D**) observed OTUs index; (**E**) Goods coverage index. Different letters indicate significant differences between groups (*p* < 0.05). Differences between groups were assessed by Dunnett post hoc test after a significant one-way ANOVA (*p* < 0.05). NC: Normal control group, chow diet; BC: Blank control group, high-fat diet; ORL: Positive control group, orlistat; TRL: Low-dose group, trilobatin; TRM: Middle-dose group, trilobatin; TRH: High-dose group, trilobatin.

**Figure 2 nutrients-13-00891-f002:**
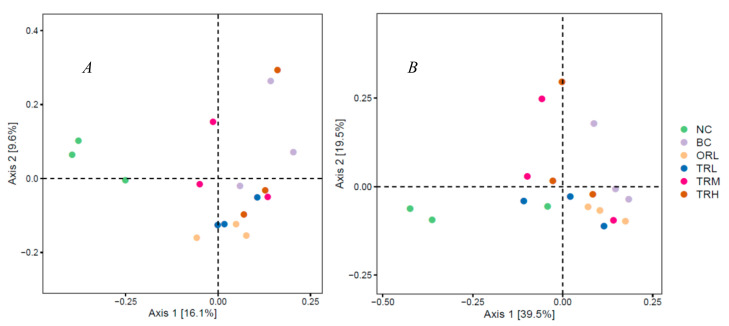
Principal coordinate analysis (PCoA) illustrating the different tested groups of NC, BC, ORL, TRL, TRM, and TRH based on unweighted (**A**) and weighted (**B**) UniFrac distances. PERMANOVAs were accessed by the Adonis method and weighted or unweighted UniFrac distance matrices to analyze significant differences in the gut microbiota composition. NC: Normal control group, chow diet; BC: Blank control group, high-fat diet; ORL: Positive control group, orlistat; TRL: Low-dose group, trilobatin; TRM: Middle-dose group, trilobatin; TRH: High-dose group, trilobatin.

**Figure 3 nutrients-13-00891-f003:**
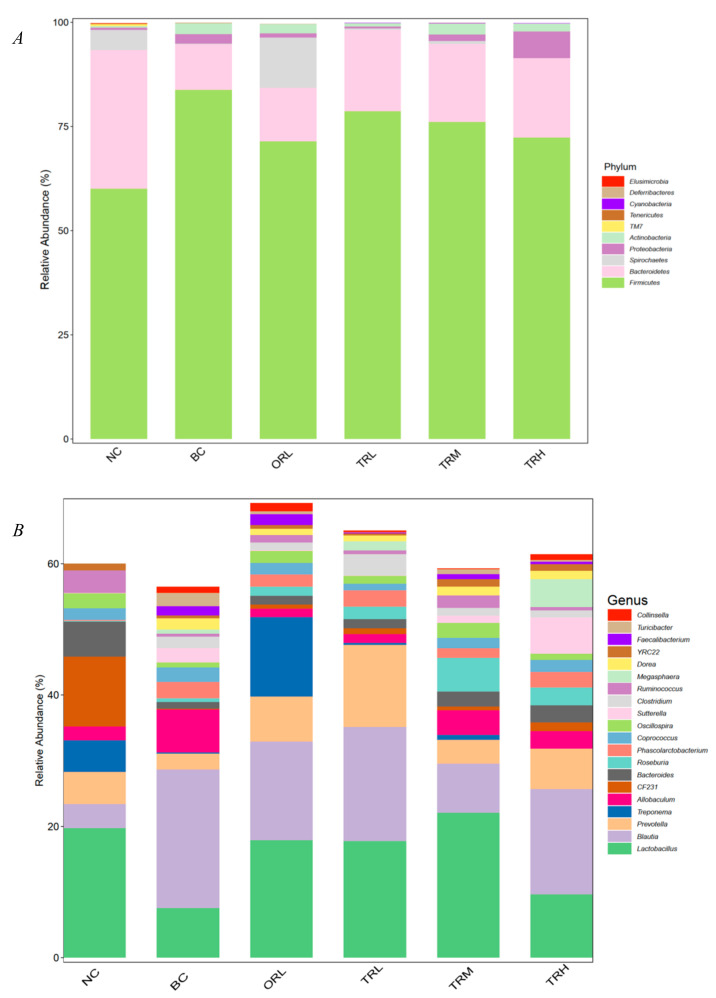
Microbiota compositions in BC, NC, ORL, TRL, TRM, and TRH groups. (**A**) Microbiota at the phylum level; (**B**) microbiota at the genus level; (**C**–**M**) relative abundances of microbiota at genus level. Different letters indicate significant differences between groups (*p* < 0.05). Differences between groups were assessed by Dunnett post hoc test after a significant one-way ANOVA (*p* < 0.05). NC: Normal control group, chow diet; BC: Blank control group, high-fat diet; ORL: Positive control group, orlistat; TRL: Low-dose group, trilobatin; TRM: Middle-dose group, trilobatin; TRH: High-dose group, trilobatin.

**Figure 4 nutrients-13-00891-f004:**
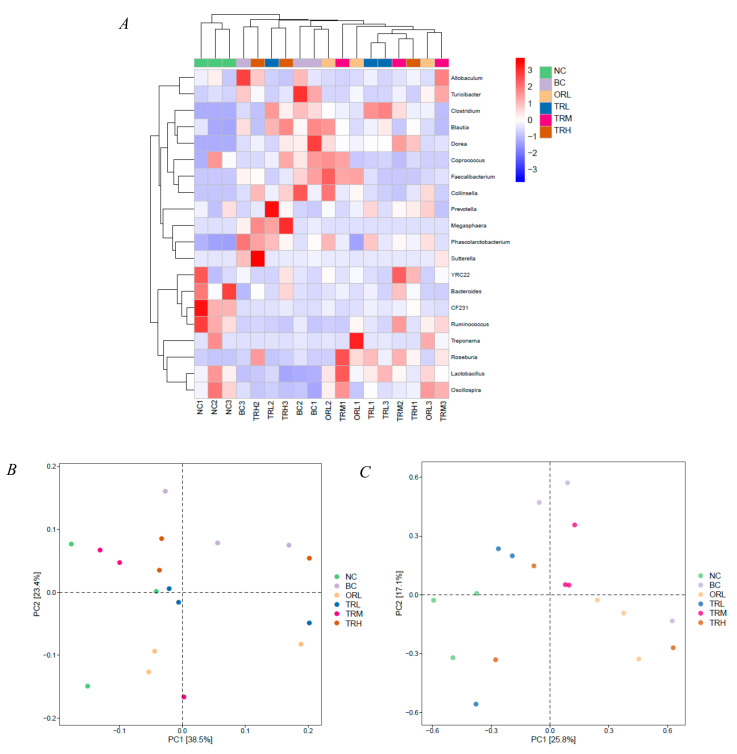
Heatplot (**A**), PCA (**B**), and OPLS-DA (**C**) analysis of the abundance of gut microbiota at genus level in the tested groups. NC: Normal control group, chow diet; BC: Blank control group, high-fat diet; ORL: Positive control group, orlistat; TRL: Low-dose group, trilobatin; TRM: Middle-dose group, trilobatin; TRH: High-dose group, trilobatin.

**Figure 5 nutrients-13-00891-f005:**
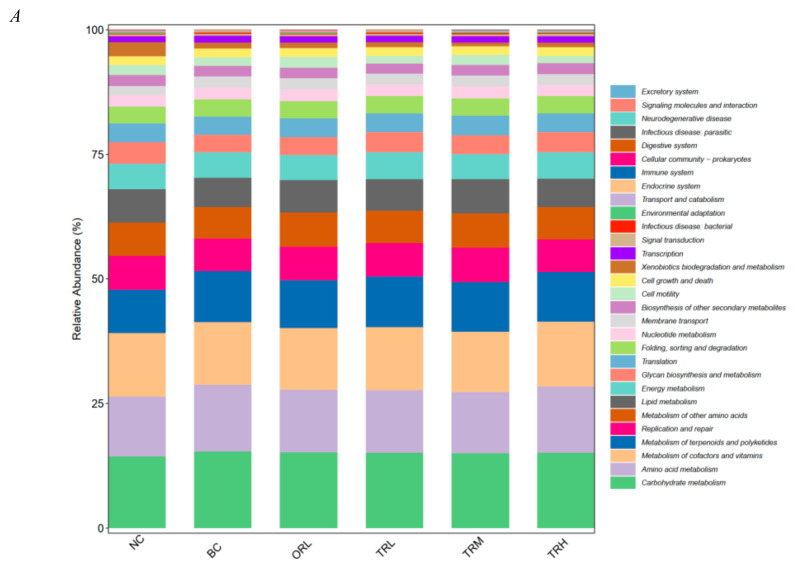
KEGG pathways prediction (**A**) and correlations between the microbiome at genus level and metabolites (**B**) in different groups. NC: Normal control group, chow diet; BC: Blank control group, high-fat diet; ORL: Positive control group, orlistat; TRL: Low-dose group, trilobatin; TRM: Middle-dose group, trilobatin; TRH: High-dose group, trilobatin.

**Table 1 nutrients-13-00891-t001:** Food intake and morphometric parameters at the sacrifice day for rats in the chow diet, high-fat diet, and trilobatin and orlistat treatment groups.

		NC	BC	ORL	TRL	TRM	TRH
Food intake	Sum total intake (g)	124.72 ± 4.57 a	135.08 ± 2.09 a	130.99 ± 6.65 a	126.71 ± 5.86 a	125.20 ± 10.2 a	126.11 ± 3.02 a
Morphometric parameters	Final body weight (g)	453 ± 11.4 d	492.42 ± 10.29 a	467.08 ± 12.9 cd	472.50 ± 11.26 b	457.75 ± 10.4 cd	475.58 ± 9.57 b
	Gain body weight (g)	59.17 ± 3.50 cd	85.92 ± 4.07 a	66.58 ± 8.00 bc	71.50 ± 6.78 b	56.17 ± 5.09 d	74.75 ± 5.35 b
	Liver weight (g)	12.35 ± 1.23 c	17.67 ± 1.11 a	16.18 ± 1.21 b	16.63 ± 1.28 bc	16.43 ± 1.2 bc	16.40 ± 1.41 bc
	PAT (g)	3.59 ± 0.46 c	8.01 ± 1.01 a	5.19 ± 0.42 bc	6.19 ± 0.41 b	5.20 ± 0.47 bc	5.07 ± 0.49 bc
	EAT (g)	4.06 ± 0.16 b	5.23 ± 0.36 a	5.07 ± 0.41 ab	5.03 ± 0.19 ab	4.30 ± 0.32 b	4.52 ± 0.27 b
	BAT (g)	0.47 ± 0.06 c	0.68 ± 0.11 a	0.57 ± 0.09 b	0.61 ± 0.11 ab	0.59 ± 0.09 ab	0.55 ± 0.07 ab
	Sum AT (g)	8.12 ± 0.13 c	13.92 ± 0.57 a	10.83 ± 0.54 b	11.83 ± 0.19 bc	10.09 ± 0.25 b	10.14 ± 0.23 b
Short-chain fatty acids (ng/μL)	Acetic acid	5.11 ± 0.25 a	2.49 ± 0.11 b	0.67 ± 0.03 c	4.65 ± 0.20 ab	2.37 ± 0.12 bc	1.66 ± 0.09 bc
	Propionic acid	4.89 ± 0.31 b	4.54 ± 0.21 b	1.55 ± 0.12 c	3.85 ± 0.18 bc	6.17 ± 0.25 a	2.47 ± 0.11 bc
	Isobutyric acid	5.23 ± 0.54 a	3.18 ± 0.39 a	4.19 ± 0.46 a	3.54 ± 0.61 a	3.30 ± 0.87 a	4.32 ± 0.35 a
	Butyric acid	3.37 ± 0.46 b	2.52 ± 0.13 c	5.44 ± 0.45 a	2.68 ± 0.18 c	2.51 ± 0.16 c	5.29 ± 1.05 a
	Isovaleric acid	5.95 ± 0.66 a	6.04 ± 1.01 a	2.23 ± 0.11 a	5.23 ± 0.56 a	3.62 ± 0.23 a	2.61 ± 0.31 a
	Valeric acid	4.79 ± 0.35 a	4.97 ± 0.23 a	2.95 ± 0.12 a	4.50 ± 0.55 a	3.19 ± 0.42 a	2.99 ± 0.36 a

Values are represented as mean ± standard deviations. Values in a row with the same letter are not significantly different (*p* > 0.05). Difference between groups was assessed by Dunnett post hoc test after a significant one-way ANOVA (*p* < 0.05). NC: Normal control group, chow diet; BC: Blank control group, high-fat diet; ORL: Positive control group, orlistat; TRL: Low-dose group, trilobatin; TRM: Middle-dose group, trilobatin; TRH: High-dose group, trilobatin; PAT: Perirenal adipose tissue; EAT: Epididymal adipose tissue; BAT: Brown adipose tissue; AT: Adipose tissue.

## Data Availability

Data available on reasonable request from the authors.

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
