# Peer review of "Effect of Trilobatin from Lithocarpus polystachyus Rehd on Gut Microbiota of Obese Rats Induced by a High-Fat Diet"

_nutrients, 2021, doi:10.3390/nu13030891_

Round 1
Reviewer 1 Report
Authors of the manuscript entitled „Effect of Trilobatin from Lithocarpus Polystachyus Rehd on the Serum Lipid Levels and Gut Microbiota of Rats under High-fat Diet” present an interesting topic in the field of excess body weight and natural plant extracts, but major corrections are necessary.
General
In my opinion, Authors present too many results. It would be better to focus on one aspect, for example, the influence of the Trilobatin from Lithocarpus Polystachyus Rehd on lipid levels and metabolism, or the influence of the Trilobatin on gut microbiota of rats. In the present form the manuscript is very long and difficult to read, moreover the presented results are hardly readable.
Introduction
- The Introduction requires major corrections. In the current version, it is not a sufficient introduction for the presented research topic. The first part of the introduction does not correspond to the title and the results presented in the manuscript (this study was not conducted in obese rats).
- Authors wrote that “ However, the dieting and strenuous exercise require the long-term persistence and are easy to rebound, while drug therapy used for anti-obesity costs highly and usually has serious side effects. Thus, the plant extracts and compounds used as anti-obesity have been attracted the researchers for diet supplement due to the safety and no negative side effects. This statement suggests that the use of dietary supplements is more effective than diet therapy and physical activity, but this is not true.
- I do not agree with the statement that plant extracts are safe and do not cause side effects, because they can interact with medications or food ingredients, cause adverse reactions, for example, patients with hypertension cannot consume licorice, which causes an increase in blood pressure.
- The Introduction does not correspond to the title of the manuscript, because Authors do not present the issue related to the level of lipids and their metabolism.
- The aim of the study stated by Authors at the end of this section does not refer to the level of lipids.
Materials and methods
- How old were the rats? Please complete this information.
- How did Authors determine the dose levels of Trilobatin?
- One group of animals received Orlistat – please clarify it.
- Please explain in Discussion section how the method of administration of trilobatin (intragastrically) might have influenced the obtained results.
- Authors write about liver removed, so please explain in this section what laboratory assay were performed in the liver.
- Authors should complete the information about the statistical tests that have been applied. How did Authors verify the normality of distribution of data? What tests did they use?
- Please change the marking of statistically significant differences (a,b,c) in all tables and figures.
- There is no information about the analysis of the metabolic pathways.
Results
- Figure 2, Figure 3 and Figure 6 are illegible.
- Description under Figure 3, Figure 4, Figure 6 are incomprehensible.
- Section 3.5 – please present the correlations in the table.
Discussion
- Authors report that excess fat in the diet causes obesity, but they forget that the excess of simple sugars also contributes to the development of obesity.
- How is it possible to explain the fact that only the average dose of Trilobatin influenced the fat content in rats?
- Authors cannot write that "It was noted that trilobatin from LPR leaves supplementation could dramatically reversed the high-fat diet induced increase in TG”, because they did not evaluate the effect of Trilobatin on TG levels in animals with high TG levels.
- Please explain what could be the reason for the influence of the average dose of Trilobatin on the TG level in serum?
- Authors refer the results obtained in this study to obese people, however, they did not conduct the study in obese rats. This way of interpreting the results may mislead the reader.
- If Trilobatin caused a significant inhibition of weight gain in non-obese rats, what can be the conclusion? Does this mean that people with normal body weight can use this extract safely?
- Authors wrote: „The food intake during 4-week treatment among all tested groups was not significantly (p > 0.05) compared with high-fat diet group, which was demonstrated the trilobatin induced weight loss was involved a mechanism that was independent of hyperphagia” Please clarify it.
- Please do not use wording such as: „4-week trilobatin supplementation reversed the levels of TC, TG and LDL-C of rats induced by high-fat diet, especially TC and LDL-C”, because the groups of rats were compared at one time point, changes at two time points were not compared (baseline vs after 4 weeks of using the extract).
- Please indicate possible side effects of administration of Trilobatin extract isolated from leaves.
- Please explain how the results can be applied in humans in terms of public health? Did Authors suggest using the extract in the prevention or treatment of obesity?
- Authors should present the limitations of their study.
- Please specify information about the possible future research directions.
References
References must be corrected.
- Authors must complete the DOI for their references.
- Authors should follow the instructions for authors (the way of referring is incorrect).
Author Response
We thank the referees for his/her extensive and detailed report. We do believe that the questions and criticisms will help us to improve the quality and clearness of manuscript. We will try to answer his/her report point by point. We have revised the sentences in manuscript by the red color.

Reviewer 2 Report
In the paper “Effect of Trilobatin from Lithocarpus Polystachyus on the Serum Lipid Levels and Gut Microbiota of Rats under High-fat Diet.” Authors have demonstrated that trilobatin could significantly prevent high-fat diet induced body and liver weight gain, and the accumulation of perirenal fat, epididymal fat and brown fat of SD rats in a dose-independent manner. Author also showed that Short chain fatty acids (SCFA) concentrations increased and increase of triglyceride (TG), total cholesterol (TC) and low-density lipoprotein cholesterol levels induced by high-fat diet was attenuated by trilobatin supplementation. This work required minor revision. I recommend the authors to address the points in the revision of the paper. My comments are below.
- In abstract section, correct does-independent manner to dose-dependent manner.
- How did author isolate the short chain fatty acids? Please describe in a separate paragraph under Materials and methods section.
- What is the y-axis in Figure 1? Include in the figure.
- Include A, B and C in the Figure 2 and cite Figure 2A, 2B etc. wherever necessary in body text of the manuscript.
- Include D and E in Figure 4 and A in Figure 6.
- All figures are not systematically arranged. Arrange figure properly with sub figures A, B, C, etc. and change legends accordingly and also cite properly in the text body of manuscript.
Author Response
We thank the referees for his/her extensive and detailed report. We do believe that the questions and criticisms will help us to improve the quality and clearness of manuscript. We will try to answer his/her report point by point. We have revised the sentences in manuscript by the red color and the detailed modifications are in the attached Word file.

Round 2
Reviewer 1 Report
Unfortunately, Authors of the manuscript entitled „Effect of Trilobatin from Lithocarpus Polystachyus Rehd on Gut Microbiota of obese rats induced by a high-fat diet ” did not introduce changes to the manuscript in accordance with previous suggestions.
- My previous comment: "The first part of the introduction does not correspond to the title and the results presented in the manuscript." Authors did not correct this Section.
- My previous comment: "Authors wrote that “However, the dieting and strenuous exercise require the long-term persistence and are easy to rebound, while drug therapy used for anti-obesity costs highly and usually has serious side effects. Thus, the plant extracts and compounds used as anti-obesity have been attracted the researchers for diet supplement due to the safety and no negative side effects. This statement suggests that the use of dietary supplements is more effective than diet therapy and physical activity, but this is not true. I do not agree with the statement that plant extracts are safe and do not cause side effects, because they can interact with medications or food ingredients, cause adverse reactions, for example, patients with hypertension cannot consume licorice, which causes an increase in blood pressure." Authors did not correct this Section.
- My previous comment: "
Authors should complete the information about the statistical tests that have been applied. How did Authors verify the normality of distribution of data? What tests did they use?" Authors did not do it.
- Descritptions are still incomprehensible.
- The marking (*, #) used by the Authors are illegible.
My previous comment: "Authors wrote: „The food intake during 4-week treatment among all tested groups was not significantly (p > 0.05) compared with high-fat diet group, which was demonstrated the trilobatin induced weight loss was involved a mechanism that was independent of hyperphagia” Please clarify it." The explanation is insufficient.
My previous comment: "
Please indicate possible side effects of administration of trilobatin extract isolated from leaves. Please explain how the results can be applied in humans in terms of public health? Did Authors suggest using the extract in the prevention or treatment of obesity? Authors should present the limitations of their study. Please specify information about the possible future research directions." The explanation is insufficient.
Authors did not significantly correct the manuscript.
Author Response
Answer to the reviewers’ report of manuscript
Title: Effect of Trilobatin from Lithocarpus Polystachyus Rehd on Gut Microbiota of obese rats induced by a high-fat diet
We thank the referees for his/her extensive and detailed report. We do believe that the questions and criticisms will help us to improve the quality and clearness of manuscript. We will try to answer his/her report point by point. We have revised the sentences in manuscript by the red color.
Suggestions:
- My previous comment: "The first part of the introduction does not correspond to the title and the results presented in the manuscript." Authors did not correct this Section.
Thank you very much for reminding, we added one paragraph in the part of introduction to explain the important of gut microbiota for lipid metabolism at line 62-73.
- My previous comment: "Authors wrote that “However, the dieting and strenuous exercise require the long-term persistence and are easy to rebound, while drug therapy used for anti-obesity costs highly and usually has serious side effects. Thus, the plant extracts and compounds used as anti-obesity have been attracted the researchers for diet supplement due to the safety and no negative side effects. This statement suggests that the use of dietary supplements is more effective than diet therapy and physical activity, but this is not true. I do not agree with the statement that plant extracts are safe and do not cause side effects, because they can interact with medications or food ingredients, cause adverse reactions, for example, patients with hypertension cannot consume licorice, which causes an increase in blood pressure." Authors did not correct this Section.
Yes, it is true, we used “However, the dieting and strenuous exercise usually require the long-term persistence, while drug therapy for anti-obesity are appetite suppressants by regulating neurotransmitters in the hypothalamus, but it has a lots limitations including the side effects, high recurrence rate, etc [6]. Thus, finding the natural plant extracts and compounds to cure obesity is very important.” Instead of the previous sentence at line 42-46. ”
- Authors should complete the information about the statistical tests that have been applied. How did Authors verify the normality of distribution of data? What tests did they use?" Authors did not do it.
We used the “Kolmogorov-Smirnov test” to verify the normality of data distribution, and add the information at line 139-143.
- The marking (*, #) used by the Authors are illegible.
According to your suggestions, we used the a, b, c etc, to indicate the significance in the table 1, figure 1 and 3.
- Authors wrote: The food intake during 4-week treatment among all tested groups was not significantly (p > 0.05) compared with high-fat diet group, which was demonstrated the trilobatin induced weight loss was involved a mechanism that was independent of hyperphagia” Please clarify it." The explanation is insufficient.
We used “Compared to the BC group, the food intake during 4-week trilobatin treatment was not significant (p > 0.05), which indicated the trilobatin induced weight loss of obese rats may be due to the abnormal absorption of foods or the decrease of body energy storage in vivo [14, 24].” to instead of the previous sentence at line 265-268.
- Please indicate possible side effects of administration of trilobatin extract isolated from leaves. Please explain how the results can be applied in humans in terms of public health? Did Authors suggest using the extract in the prevention or treatment of obesity? Authors should present the limitations of their study. Please specify information about the possible future research directions." The explanation is insufficient.
According to your suggestions, we added the future perspectives of our researches at line 323-332.
